# The Validity and Predictive Value of Blood-Based Biomarkers in Prediction of Response in the Treatment of Metastatic Non-Small Cell Lung Cancer: A Systematic Review

**DOI:** 10.3390/cancers12051120

**Published:** 2020-04-30

**Authors:** Frederik van Delft, Hendrik Koffijberg, Valesca Retèl, Michel van den Heuvel, Maarten IJzerman

**Affiliations:** 1Health Technology and Services Research Department, Technical Medical Centre, University of Twente, Hallenweg 5, 7522 NH Enschede, The Netherlands; f.a.vandelft@utwente.nl (F.v.D.); h.koffijberg@utwente.nl (H.K.); v.retel@nki.nl (V.R.); 2Division of Psychosocial Research and Epidemiology, Netherlands Cancer Institute, Plesmanlaan 121, 1066 CX Amsterdam, The Netherlands; 3Respiratory Diseases, Radboud University Medical Center, Geert Grooteplein Zuid 10, 6525 GA Nijmegen, The Netherlands; Michel.vandenHeuvel@radboudumc.nl; 4Centre for Cancer Research and Centre for Health Policy, University of Melbourne, Parkville, VIC 3000, Australia; 5Peter MacCallum Cancer Centre, Parkville, VIC 3000, Australia

**Keywords:** liquid biopsy, non-small cell lung cancer, biomarkers

## Abstract

With the introduction of targeted therapies and immunotherapy, molecular diagnostics gained a more profound role in the management of non-small cell lung cancer (NSCLC). This study aimed to systematically search for studies reporting on the use of liquid biopsies (LB), the correlation between LBs and tissue biopsies, and finally the predictive value in the management of NSCLC. A systematic literature search was performed, including results published after 1 January 2014. Articles studying the predictive value or validity of a LB were included. The search (up to 1 September 2019) retrieved 1704 articles, 1323 articles were excluded after title and abstract screening. Remaining articles were assessed for eligibility by full-text review. After full-text review, 64 articles investigating the predictive value and 78 articles describing the validity were included. The majority of studies investigated the predictive value of LBs in relation to therapies targeting the epidermal growth factor receptor (EGFR) or anaplastic lymphoma kinase (ALK) receptor (n = 38). Of studies describing the validity of a biomarker, 55 articles report on one or more EGFR mutations. Although a variety of blood-based biomarkers are currently under investigation, most studies evaluated the validity of LBs to determine EGFR mutation status and the subsequent targeting of EGFR tyrosine kinase inhibitors based on the mutation status found in LBs of NSCLC patients.

## 1. Introduction

Lung cancer is the leading cause of cancer-related deaths worldwide and is known for its high incidence and mortality rates [1,2]. The current treatment standard for early-stage non-small cell lung cancer (NSCLC) (stage I–II) is resection. In addition to resection, patients are offered stereotactic radiotherapy, adjuvant chemotherapy, or a combination of both depending on the tumor stage. Patients diagnosed with stage III–IV are eligible for systemic therapy such as chemotherapy, chemoradiotherapy, immunotherapy, combinations, and targeted therapy based on the presence of specific genetic druggable mutations [3].

With the introduction of targeted therapies and immunotherapy, molecular testing has become more important, as the effectiveness of selected therapies depends on the presence of specific molecular or genomic alterations [4]. Furthermore, patients tend to develop treatment resistance leading to disease progression and requiring the use of second- and third-line targeted therapies. Moreover, tumor heterogeneity and clonal evolution possibly play a role in the development of drug resistance, increasing the need for repeat biopsies to guide treatment decisions [5,6,7]. Currently, oncogenic mutations are derived from tumor tissue samples obtained by means of invasive biopsies. However, (re-)biopsies can be highly complex and sometimes require an invasive procedure, therefore unfeasible in a substantial proportion of patients, due to location, tumor size, or general health status. Besides being performed in case of an inadequate first sample, re-biopsies can be performed to track tumor progression and clonal evolution. Since a re-biopsy does also not always provide sufficient tissue for molecular testing, liquid biopsies (LBs) might help overcome this issue [8,9,10]. LBs provide an alternative way of obtaining genetic information without the need for an invasive procedure, the low patient burden allows for more frequent biopsies to guide treatment decisions. LBs are also expected to have a health economic benefit by better treatment targeting and by earlier identification of non-response [11].

Liquid biopsies obtained through blood samples contain different types of tumor-related genetic or protein markers, e.g., circulating tumor cells (CTCs), RNA, exosomes, carcinoembryonic antigen, cytokeratin, and cell-free DNA (cfDNA), which can provide valuable information with regard to prognostics, early disease detection, treatment response monitoring, identification of emerging treatment resistance, and recurrence monitoring [12,13].

Despite the potential benefits of liquid biopsies, not many LBs are routinely used nor are they reimbursed. This partly has to do with the challenges of genome backward development of new biomarkers compared to biomarker and drug co-development, as the evidence of clinical validity needs to be established post-hoc for the latter. While clinical utility to change patient management is the holy grail in such studies, an essential and intermediate requirement is to demonstrate diagnostic validity and predictive validity. Existing reviews focus on a specific mutational pathway or biomarker and not on the evidence of validity obtained through prospective studies. This review aims to identify papers investigating a rage of liquid biopsy-based biomarkers for patients with NSCLC and, second, to extract and compare evidence of clinical and predictive validity. It primarily focuses on the diagnostic and predictive validity, where diagnostic validity is defined as the ability of LBs to identify mutations or biomarkers for which tissue analysis currently is the gold standard. Predictive validity on the other hand is defined as if the LB can predict response to a particular treatment, which is a requirement for a test to have utility.

## 2. Results

The initial search in Scopus (*n* = 1489) and PubMed (*n* = 1037) returned a total of 2526 studies, and after duplicate removal, 1704 unique records were identified. Based on the screening of all abstracts of the 1704 identified records, 1323 records (78%) were excluded from the full-text review. The full study selection process is depicted in Figure 1, and a detailed description is provided in Table 1 and Table 2 for presenting the full reference to the selected studies.

### 2.1. Extracted Data on the Validity of the Biomarker

A description of all studies included in the validity group is provided in Table 1.

Figure 2 depicts the reported sensitivity, specificity, and concordance rate of the different biomarkers identified. Results were stratified according to the analysis method used. For each study, the sensitivity, specificity, and concordance rate of LBs compared to tissue biopsies (TBs) were extracted from the included papers. Data were only extracted if the study had included at least 10 patients for whom the biomarker was also detected in a matched tissue sample. The number of included patients ranged from 10 to 989 with a mean of 55 patients.

The majority of studies (72%, *n* = 56) reported validity of EGFR mutations, including exon 19 deletion, L858R, and T790M mutations. Reported sensitivity values for identified biomarkers ranged from 19.6% to a perfect 100%. In these studies, the sensitivity was reported for EGFR, exon 19 deletion, L858R, and T790M in 23, 21, 23, and 10 studies. The results indicate that next generation sequencing (NGS) is more sensitive than polymerase chain reaction (PCR) in the detection of EGFR and T790M mutations, but less for L858R mutations. Figure 2 depicts the sensitivity, specificity, and concordance reported by each. As shown in Figure 2, the average sensitivity of NGS in the detection of EGFR and T790M mutations was 81% and 87%, respectively. While the average sensitivity of PCR in the detection of EGFR and T790M mutations was 62% and 64%, respectively. A slightly higher sensitivity of PCR compared to NGS was reported for exon 19 deletions (NGS 67%, PCR 76%).

Specificity was reported in 21, 20, 20, and 8 studies for L858R, exon 19 deletion, EGFR, and T790M mutations respectively. A specificity of >90% was seen in most of the studies, despite a few exceptions like a study reporting a specificity of 47% in a 50-gene panel including EGFR, ALK, and KRAS [19]. The specificity of L858R mutation detection was 97.8% and 98.2% for PCR and NGS-based methods respectively. While the average specificity for PCR- and NGS-based methods in the detection of exon 19 deletion was 98% and 97%, respectively. In the detection of T790M mutations with an average reported specificity of 94% and 82% for NGS- and PCR-based methods.

Finally, the concordance between LBs and TBs is reported as a percentage agreement. Concordance rates of EGFR mutation detection were reported in 14, 15, 14, and 6 studies for L858R, exon 19 deletion, EGFR, and T790M mutations, respectively. Concordance ranged from 40% for detection of the T790M mutation to 98.7% for the detection of EGFR mutations. On average reported concordance rates were higher for NGS-based methods compared to PCR-based methods for all EGFR mutations. With an average concordance rate for NGS and PCR of 91% vs. 88% in L858R mutations, 90% vs. 87% for exon 19 deletions, 89% vs. 84% for EGFR mutations, and 69% vs. 68% for T790M mutations.

### 2.2. Study Evidence Levels for Predictive Biomarkers

A description of all studies included as describing the predictive value of biomarkers is listed in Table 2.

Studies were classified according to the evidence framework as proposed by Rao et al. [147]. Six different evidence levels were identified, ranging from retrospective non-case/control studies, to post-hoc biomarker correlative analysis of a prospective randomized clinical trial. The majority of studies were classified as III B, a prospective observational study (*n* = 38.59%). Other classes included I D post-hoc biomarker correlative analysis of a prospective randomized controlled trial (*n* = 6.10%), II B prospective biomarker driven non-randomized clinical trial (*n* = 5.8%), II C a post-hoc biomarker correlative analysis of non-randomized clinical trial (*n* = 3.5%), III C a case-control study (*n* = 1.2%), III E a retrospective non-case-control study (*n* = 11.17%)(Figure 3.).

### 2.3. Evidence of Predictive Value of a Biomarker Based on LBs

A total of 64 studies were identified reporting on the predictive value of a LB to guide a specific treatment. The included studies tested 67 different analytes for 24 different treatments or treatment combinations. EGFR mutations (including exon 19 deletion, T790M, and L858R) were described in 18 studies (28%), while 10 studies described the predictive value of CTC count (16%). Nineteen studies (30%) evaluated the LB to indicate chemotherapy, either a single, doublet or combination therapy. Targeted therapy agents (e.g., erlotinib, gefitinib, icotinib, afatinib) were subject of evaluation in 31 (48%) of the identified studies, while immunotherapy agents (e.g., patritumab, nivolumab, bevacizumab) were described by 9 (14%) studies.

### 2.4. Evidence Level Per Analyte and Therapy

Figure 4 depicts the analytes and therapies described in the different studies, stratified according to the evidence level. As previously shown in Figure 3, the majority of studies were classified as class III B, a prospective observational study. In this category, CTC count, EGFR mutations, and cfDNA level were identified most frequently. While CTC count and cfDNA level were researched in combination with several types of treatments including chemotherapy, immunotherapy, and targeted therapies. EGFR mutations in this category were exclusively researched in combination with targeted therapies. Looking at the class with the highest evidence level (I D, post-hoc biomarker correlative analysis of a prospective randomized clinical trial), we see that majority of studies in this class evaluate EGFR mutations including exon 19 deletion and L858R.

## 3. Discussion

Our results provide a clear overview of the current developments within the field and the potential clinical utility of the biomarkers identified in our study. More specific, our findings suggest that in the diverse and active landscape of biomarker research, many studies focus on EGFR mutation detection in LBs. The review also concludes that the EGFR is a valid marker in comparison to tissue analysis. It was shown that using these LB markers it is possible to indicate the treatment likely to be effective.

Results show a significant variety in reported sensitivity, specificity, and concordance values for LB results compared to matched tissue samples. The variation in results might be explained by differences in sample preparation, sample volume, used assay, previous lines of treatment prior to study inclusion, disease stage, amount of tumor shedding, and the number of patients included in the study. The difference in sensitivity and specificity of the platform used in mutation detection was also shown in a review by Li et al. [148]. In this review, the authors compared the performance of multiple platforms in the detection of T790M mutations. In a review, Kim et al. [149] reported on the sensitivity, specificity, and concordance rate in the detection of EGFR mutations. In this review, authors reported a variation in outcomes depending on the technology and the genomic mutation of interest. Variations in the sensitivity of mutation detection might indicate that at this moment, LBs are not ready to replace TBs in practice; however, LBs might be a good alternative in patients in whom a TB was deemed unfeasible. Moreover, as shown in Figure 2, there are studies reporting sensitivity values exceeding 90%, indicating that by selecting the correct analysis method and patient group, LBs might provide a satisfactory sensitivity for clinical applications. The variation in concordance and specificity might be an indication of tumor heterogeneity missed by TB and would indicate that there might be an added value of performing LBs alongside tissue analysis. In a report, the International Association for the Study of Lung Cancer recommended the use of LB techniques to detect EGFR mutations in treatment naïve patients. However, a negative result should be considered uninformative and should be followed by a TB [150]. Moreover, the dominant presence of studies reporting on the clinical validity of LBs in the detection of EGFR mutations found in this review is in line with the view of the International Association for the Study of Lung Cancer, and it is to be expected that the first role of LBs in the management of NSCLC will involve detection of EGFR mutations. Our results are potentially biased towards the evaluation of the validity of LBs in the detection of EGFR mutations. Considering the potential of LBs in the detection of acquired resistance to 1st and 2nd generation EGFR tyrosine kinase inhibitors (TKIs) attracted considerable attention; however, the introduction of osimertinib, a 3rd generation EGFR TKI, lessens the need for detection of the T790M resistance mechanism, for which the FDA approved the use of plasma ctDNA analysis. Moreover, current guidelines now recommend the use of osimertinib in the first-line setting, further reducing the need for the detection of acquired resistance to 1st and 2nd generation EGFR TKIs [151]. Although the necessity for LBs in the detection of T790M mutations was diminished by the introduction of osimertinib, a more comprehensive frame of reference seems appropriate, since only 12% to 45% of NSCLC patients present with EGFR mutations, depending on geography, histology, and smoking status [152,153]. While more driver mutations, targetable pathways, drugs, and companion diagnostics are being discovered [154]. Indicating that there is a lot of potential for LBs beyond the detection of EGFR mutations and resistance mechanisms to provide clinical benefit in the future. Looking at Figure 4 it becomes apparent that a lot of biomarkers are being investigated at this moment in relation to a large variety of treatments and treatment combinations. This indicates that this is an active field in which multiple research groups try to identify the most beneficial treatment for patients based on genomic mutations or other biomarkers identified by LBs. In this review, we looked at studies reporting on treatment outcomes based on biomarker analysis prior to initiation of the study related treatment. In a number of studies, patients did receive previous lines of treatment before study inclusion. Response monitoring could also be considered a predictive value of LBs; however, response monitoring was not taken into account in this review.

Currently, most targeted treatments requiring a companion diagnostic focused on tissue-based analyses for treatment selection, as indicated by Bernabé et al. [155] and also supported by the classification of studies according to their evidence level in this review (Figure 3 and Figure 4). The preliminary nature of the evidence makes it difficult to access the clinical benefit of mutation detection using LBs since the beneficial effect of the treatment is unclear in tissue negative, plasma positive patients. Therefore, more studies should aim to include LB analysis in the study design to build on the currently available preliminary evidence. In our review, we found that 59% of identified studies were of prospective observational nature, while only 10% of the identified studies reported on a randomized clinical trial with post-hoc biomarker analysis. Future directions towards implementation might include large registry studies, which include matched tissue and LB results, and repeated LB measurements to possibly evaluate the predictive value of a LB in response monitoring.

Like every review, this review has potential limitations. Despite the generally accepted problems of selection of studies, A more fundamental problem might be the decision not to report the specific methodology used in sample preparation and analysis. This was deliberately chosen as our focus was to review evidence levels for each of the analytes. However, it is acknowledged that specific analytic issues (such as DNA extraction) will potentially impact the clinical validity and predictive value. One of the reasons for this restriction, was that more than half of the included studies in the validity group did not provide detailed information regarding the applied methodology (e.g., DNA input quantity), referred back to previous work, only listed the test kits used, the authors stated that DNA purification or library preparation was performed according to manufacturers’ instructions, or sample analysis was performed in an external laboratory. This lack of information makes it difficult to compare different test accuracies, even within biomarkers analyzed using similar methodologies (e.g., NGS or PCR). Second, TBs are regarded as the gold standard in determining the sensitivity, specificity, and concordance rate of LBs. In this review we did not collect information on the methods used to detect biomarkers in tissue samples, the accuracy of methodologies used in the analysis of tissue samples directly influences the accuracy of LB results, e.g., mutations missed in the analysis of tissue samples potentially lead to a reduction in the specificity and concordance rate of the LB analysis. However, it was expected that all studies included in this review applied generally accepted methods or used commercially available equipment in the analysis of tissue samples.

## 4. Materials and Methods

### 4.1. Eligibility Criteria

Studies included in this review could cover a wide range of LBs, but had to present results of either the clinical validity or predictive validity. Original full-text articles published in English were selected for review.

### 4.2. Search

A systematic literature search was performed in September 2019 using Scopus and PubMed databases to identify relevant studies published between 1 January 2014 and 1 September, 2019. The time span was selected to cover all recent developments in LB development while maintaining a amenable amount of search results. The search included the following keywords and allowed for different conjugations: NSCLC, non-small cell, ctDNA, microRNA (miRNAs), CTCs, extracellular vesicles, blood, and serum. The full search queries used to perform the literature search are depicted in Appendix A. All article types were included in the initial search. This systematic review was conducted according to the Preferred Reporting Items for Systematic Reviews and Meta-Analysis (PRISMA) guidelines [156].

### 4.3. Study Selection

After removing duplicate records, a review protocol (Figure 1) was used towards the selection of relevant articles. Prior to conducting a full-text review, one author (F.v.D.) reviewed the title and abstract of all records to determine their relevance. Exclusion of records from full-text review was based on article type (e.g., review, letter to the editor, short communication, meta-analysis), cancer type (other than NSCLC), number of patients included in the study (<20), clinical utility (the absence of a relation between the biomarker and treatment outcome or a comparison between the LB results and matched tissue samples), and biomarker type (single nucleotide polymorphisms (SNPs) were excluded from this review). Inclusion criteria were checked in a fixed order (as depicted in Figure 1), inclusion criteria were not mutually exclusive and exclusion of articles was based on the first unmet criteria. All reviewed abstracts were discussed with two co-authors (V.R. and H.K.) in case of doubt, until a consensus was made on the inclusion of the paper. The two co-authors independently reviewed 70 randomly selected abstracts (~ 4% of all records identified). Results were compared to check for disagreement between reviewers.

Articles were excluded if (1) the described study included less than 20 patients, (2) the intended use of the biomarker was not categorized in terms of being prognostic, predictive, or diagnostic, (3) the study did not report overall survival (OS), progression free survival (PFS), sensitivity, specificity, and/or concordance rate, (4) full English text was unavailable.

Studies were excluded if they only reported on a biomarker of interest that could be classified as an SNP. Reported sensitivity, specificity, and concordance were only extracted in case the study included >10 patients in whom the biomarker was detected in matched tissue samples. Thresholds were chosen to ensure a minimal evidence base.

### 4.4. Data Extraction

A full-text review was conducted on all records selected by title and abstract screening to determine the eligibility of the articles for data extraction.

Full-text articles were screened for relevant outcomes, including Overall Survival (in months or days, OS), Progression-Free Survival (PFS, in months or days), Sensitivity, Specificity, and Concordance rate (percentage of identical measurement outcomes).

Records included after full-text screening were classified into two categories, namely validity and predictive value. Articles describing a direct comparison between LB and tissue-based molecular analysis were categorized in the category validity. From these papers we extracted the sensitivity, specificity, and concordance rate. The sensitivity and specificity reflect the true positive and true negative rate, respectively. While the concordance rate should reflect the overlap between LB and TB outcomes. The category predictive value was assigned to articles describing differences in clinical outcomes from study treatments based on the presence of a biomarker detected by LB analysis.

### 4.5. Evidence Classification

To gain an insight into the stage of biomarker research, we classified the level of evidence for all articles included after full-text review and categorized as describing the predictive validity of a biomarker. For this purpose, the evidence framework as proposed by Rao et al. was adopted [147]. evidence levels were classified from level I A (high-quality meta-analysis) to level IV E (expert opinion). All records were classified by the first author (F.v.D.) and discussed with co-authors (V.R. and H.K.) in case of doubt. Records were classified according to the highest applicable evidence level.

### 4.6. Data Interpretation

Information provided by included studies was summarized to provide a comprehensible overview. Meaning, in studies classified as predictive all mentioned chemotherapy agents in single, doublet, and combination therapies were labeled as chemotherapy. The therapy of interest in the study was labeled as EGFR TKI in case the study included multiple comparable EGFR therapies, e.g., erlotinib and gefitinib without stratification of results based on the prescribed therapy. In studies describing the validity of a biomarker, the detailed description of the biomarker analysis method was reduced to the principal technique or method. The distribution and average of the reported sensitivity, specificity, and concordance values were estimated using a weighted approach based on the study size.

## 5. Conclusions

Current literature shows that the field is moving towards the use of LBs in the detection of EGFR mutations and the prescription of EGFR TKI inhibitors. Moreover, the first adoption of LBs in practice is expected to involve the detection of EGFR mutations as an addition to currently employed TBs. The currently available evidence for most analytes is limited to observational studies, and the sensitivity, specificity, and concordance rates of LBs showed a strong variation between studies. Although the diagnostic accuracy of LB compared to TB results is not perfect, it should be noted that LBs might detect mutations missed in TBs, and further research is needed to evaluate the clinical benefit of adopting LBs in practice.

## Figures and Tables

**Figure 1 cancers-12-01120-f001:**
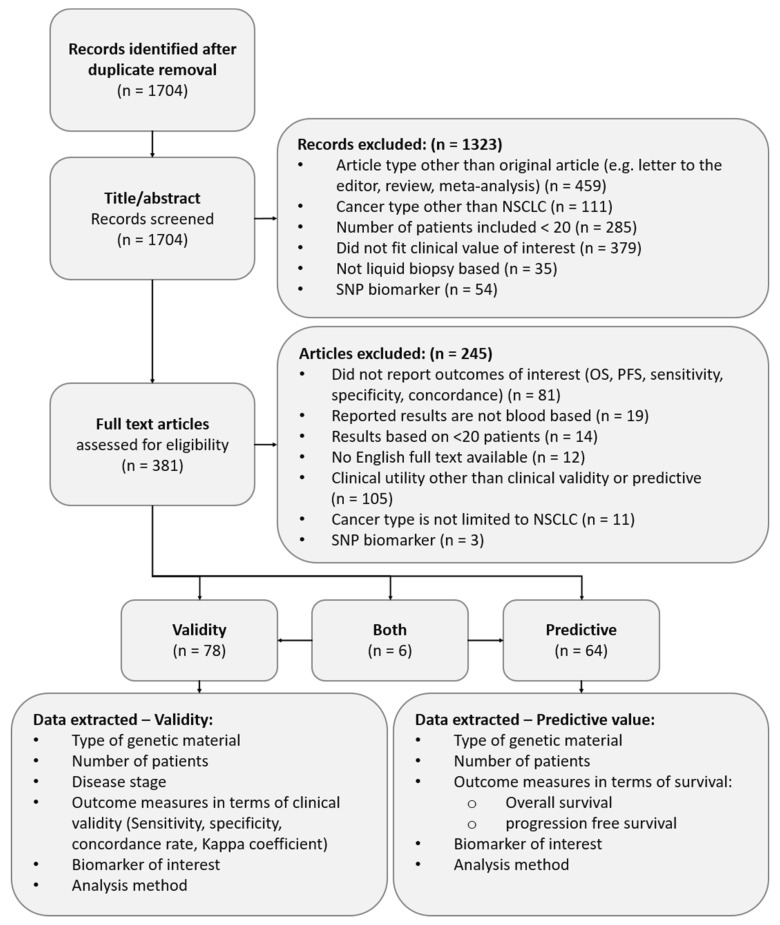
Study selection flow chart. (Non-small cell lung cancer: NSCLC, single nucleotide polymorphism: SNP, Overall survival: OS, Progression free survival: PFS).

**Figure 2 cancers-12-01120-f002:**
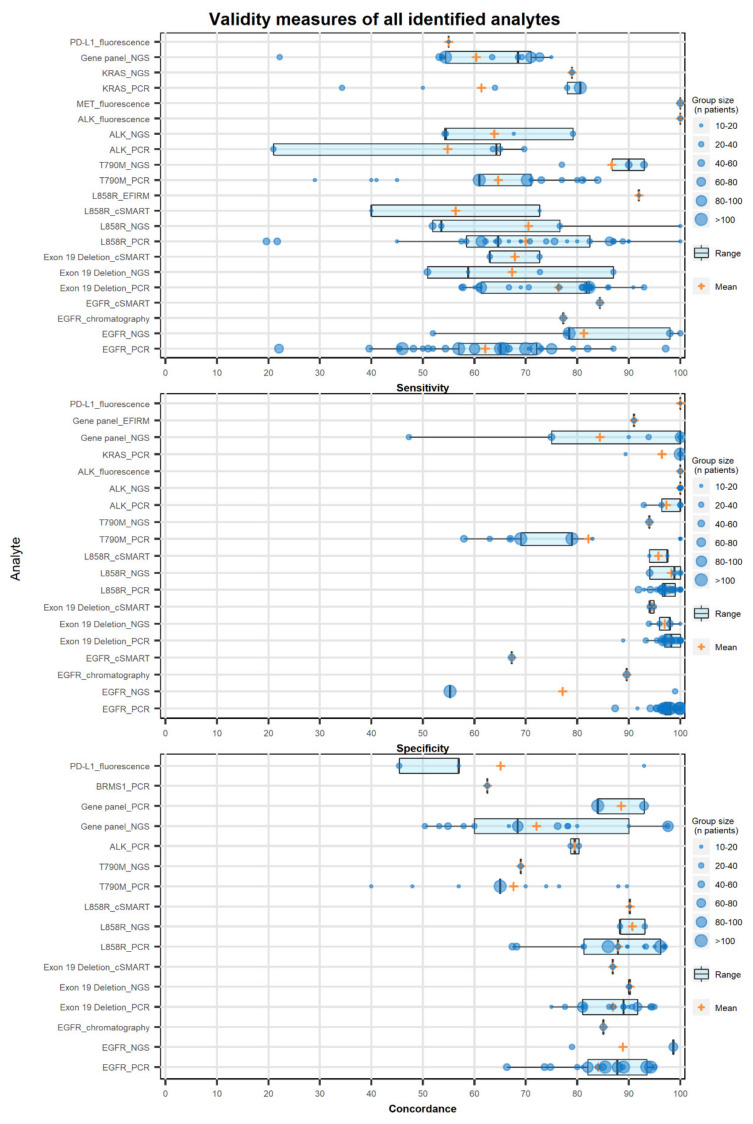
Validity measures of all identified analytes, including the sensitivity, specificity, and concordance of liquid biopsy results compared to matched tissue samples. The y-axis presents each of the reported biomarkers with analysis platform used and separated through an underscore (e.g., EGFR_NGS). The size of the “circle” (see caption right of the figure) depicts the number of patients in whom the biomarker was detected in the tissue sample. Likewise, the “plus” shaped marker depicts the average of the reported values. A boxplot is used to present the range of the reported values, the box represents the 25^th^ and 75^th^ percentiles, while the whiskers extend to a maximum of 1.5 times the inter-quartile range.

**Figure 3 cancers-12-01120-f003:**
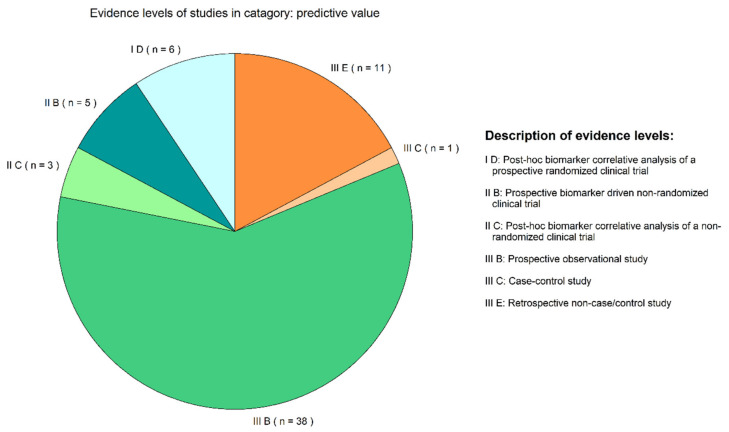
Evidence levels of studies in category: predictive value. Evidence levels identified in studies classified as describing the predictive value of liquid biopsies.

**Figure 4 cancers-12-01120-f004:**
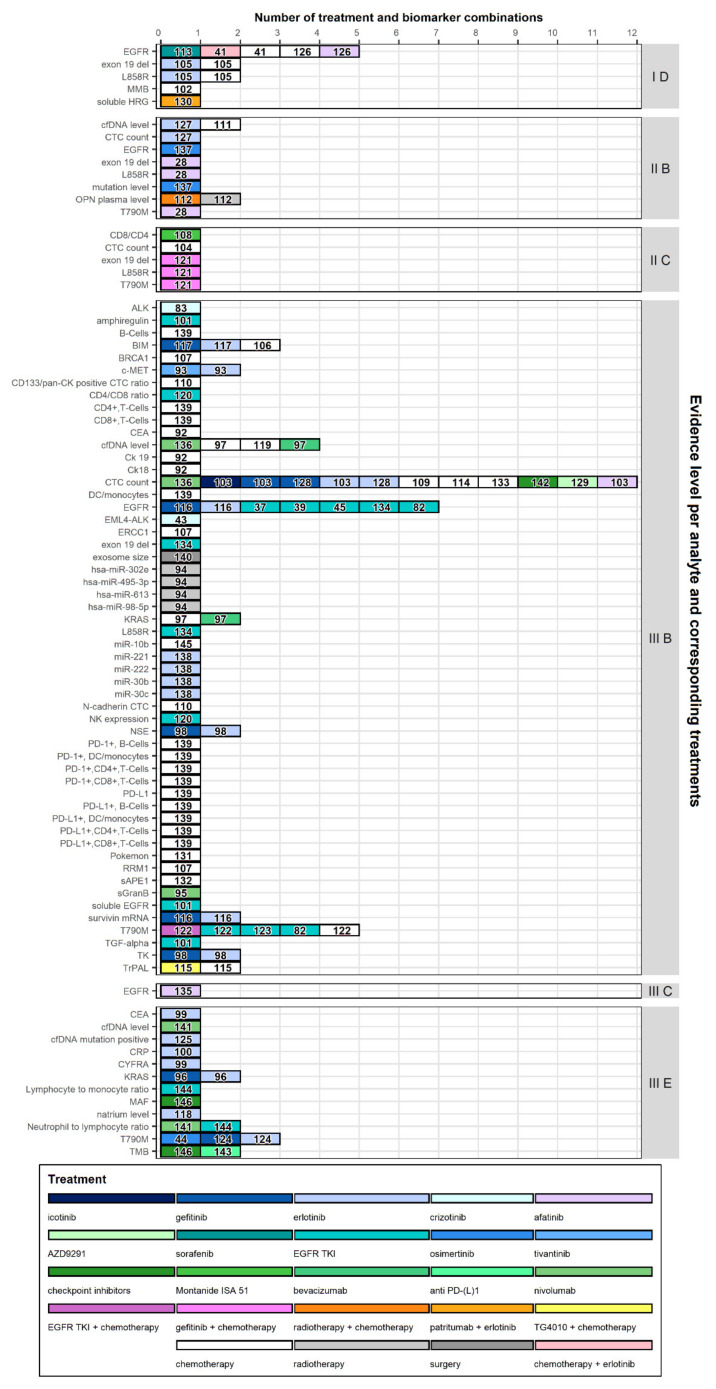
Evidence level per analyte and with reference to the companion therapies. The data is presented for each evidence level (levels I-III; right y-axis) and studies were categorized based on the biomarker of interest. Different colors are used to indicate the treatment these biomarkers were compared to and numbers within the bars refer to the corresponding reference number. The evidence levels were adopted from Rao et al. [147]. I D: Post-hoc biomarker correlative analysis of a prospective randomized clinical trial. II B: Prospective biomarker driven non-randomized clinical trial. II C: Post-hoc biomarker correlative analysis of a non-randomized clinical trial. III B: Prospective observational study. III C: Case-control study. III E: Retrospective non-case/control study.

**Table 1 cancers-12-01120-t001:** Description of included studies describing the validity of a liquid biopsy-based biomarker.

1st Author	Publication Year	Evaluated Biomarker(s)	Analysis Method	Reference Number
Alegre et al.	2016	L858R, del E746-A750	PCR	[14]
Arriola et al.	2018	EGFR	PCR	[15]
Balgkouranidou et al.	2014	BRMS1	PCR	[16]
Chai et al.	2016	Exon 19 deletion, L858R, L861Q, T790M, exon 20 insertions	cSMART	[17]
Chen et al.	2017	Gene panel	NGS	[18]
Chen et al.	2016	Gene panel	NGS	[19]
Cui et al.	2017	ALK	NGS	[20]
Douillard et al.	2014	EGFR	PCR	[21]
Duan et al.	2015	EGFR	PCR	[22]
Guibert et al.	2018	PD-L1	Fluorescence	[23]
Guibert et al.	2018	EGFR, T790M	NGS & PCR	[24]
Guibert et al.	2016	KRAS	NGS & PCR	[25]
Guo et al.	2016	Gene panel	NGS	[26]
Han et al.	2016	EGFR, KRAS	PCR	[27]
He et al.	2017	L858R, exon 19 deletion, EGFR	PCR	[28]
Ilie et al.	2018	PD-L1	Fluorescence	[29]
Ilie et al.	2017	MET	Fluorescence	[30]
Jenkins et al.	2017	T790M, L858R, exon 19 deletion	PCR	[31]
Kasahara et al.	2017	Exon 19 deletion, L858R	PCR	[32]
Krug et al.	2018	EGFR, T790M	NGS & PCR	[33]
Lam et al.	2015	EGFR	PCR	[34]
Lee et al.	2016	Exon 19 deletion, L858R	PCR	[35]
Li et al.	2017	Exon 19 deletion, L858R, gene panel	PCR	[36]
Li et al.	2014	EGFR	PCR	[37]
Liu et al.	2018	Gene panel, L858R, exon 19 deletion, KRAS, ALK	NGS	[38]
Ma	2016	EGFR, exon 19 deletion, L858R	PCR	[39]
Mayo-de-las-Casas et al.	2017	EGFR	PCR	[40]
Mok et al.	2015	EGFR, exon 19 deletion, L858R, G719X, L861Q	PCR	[41]
Muller et al.	2017	Gene panel	NGS	[42]
Nilsson et al.	2016	ALK	PCR	[43]
Oxnard et al.	2016	Exon 19 deletion, L858R, T790M	PCR	[44]
Que et al.	2016	EGFR	Chromatograpy	[45]
Reck et al.	2016	EGFR	PCR	[46]
Reckamp et al.	2016	T790M, L858R, exon 19 deletion	NGS	[47]
Sacher et al.	2016	Exon 19 deletion, L858R, T790M, KRAS	PCR	[48]
Schwaederle et al.	2017	Gene panel, EGFR	NGS	[49]
Shi et al.	2018	EGFR, exon 19 deletion, L858R	cSMART	[50]
Sim et al.	2018	EGFR	PCR	[51]
Sundaresan	2016	T790M	PCR	[52]
Sung et al.	2017	Exon 19 deletion, L858R	NGS	[53]
Tompson et al.	2016	Gene panel, EGFR	NGS	[54]
Thress et al.	2015	Exon 19 deletion, L858R, T790M	PCR	[55]
Uchida et al.	2015	L858R, EGFR, gene panel	NGS	[56]
Vazquez et al.	2016	EGFR	NGS	[57]
Wan et al.	2017	EGFR	PCR	[58]
Wang et al.	2014	EGFR	PCR	[59]
Wang et al.	2016	ALK	NGS	[60]
Watanabe et al.	2016	EGFR, exon 19 deletion	PCR	[61]
Wei et al.	2018	Gene panel, L858R, exon 19 deletion	EFIRM	[62]
Wei et al.	2017	Gene panel, L858R, exon 19 deletion	PCR	[63]
Wu et al.	2018	EGFR	PCR	[64]
Wu et al.	2017	Exon 19 deletion, L858R	PCR	[65]
Xu et al.	2016	Gene panel	NGS	[66]
Yang et al.	2017	BRAF, EGFR, exon 19 deletion, L858R, T790M	PCR	[67]
Yao et al.	2017	Gene panel	NGS	[68]
Yu et al.	2019	EGFR	PCR	[69]
Yu et al.	2017	Exon 19 deletion, L858R, T790M	PCR	[70]
Zhang et al.	2018	EGFR, L858R, exon 19 deletion	PCR	[71]
Zhang et al.	2017	L858R, exon 19 deletion	PCR	[72]
Zheng et al.	2016	T790M	PCR	[73]
Zhu et al.	2015	Exon 19 deletion, L858R	PCR	[74]
Zhu et al.	2017	Exon 19 deletion, L858R	PCR	[75]
Zhu et al.	2017	Exon 19 deletion, L858R	PCR	[76]
Chen et al.	2019	PD-L1	Fluorescence	[77]
Ding et al.	2019	Exon 19 deletion, L858R, S768I, L861Q	PCR	[78]
Garcia et al.	2019	EGFR	NGS	[79]
He et al.	2019	ALK	Fluorescence	[80]
Li et al.	2019	ALK, KRAS, EGFR, MET, ERBB2, BRAF, ROS1, RET, T790M	NGS	[81]
O’kane et al.	2019	T790M	NGS	[82]
Park et al.	2019	ALK	PCR	[83]
Soria-Comes et al.	2019	EGFR	PCR	[84]
Usui et al.	2019	T790M	NGS	[85]
Wang et al.	2019	EGFR	NGS	[86]
Yang et al.	2018	T790M	PCR	[87]
Ye et al.	2019	KRAS	PCR	[88]
Zhang et al.	2019	EGFR	NGS	[89]
Zhang et al.	2019	Exon 19 deletion, L858R	PCR	[90]
Yoshida et al.	2017	Exon 19 deletion, L858R, T790M	PCR	[91]

Polymerase chain reaction: PCR, Next generation sequencing: NGS, Circulating single molecule amplification and re-sequencing technology: cSMART, Epidermal Growth Factor Receptor: EGFR, Breast Cancer Metastasis Supressor-1: BRMS1, Anaplastic Lymphoma Kinase: ALK, Programmed Death Ligand 1: PD-L1, Kirsten Rat Sarcoma: KRAS, MET proto-oncogene: MET, B-Raf proto-oncogene: BRAF, erb-b2 receptor tyrosine kinase 2: ERBB2, ROS proto-oncogene 1: ROS1, ret proto-oncogene: RET.

**Table 2 cancers-12-01120-t002:** Description of included studies describing the predictive value of a liquid biopsy-based biomarker.

1st Author	Publication Year	Evaluated Biomarker(s)	Treatment	Reference Number
Arrieta et al.	2014	Ck18, Ck19, CEA	Chemotherapy	[92]
Azuma et al.	2016	c-MET	Erlotinib, Tivantinib	[93]
Chen et al.	2016	has-miR-98-5p, has-miR-302e, has-miR-495-3p, has-miR-613	Radiotherapy	[94]
Costantini et al.	2018	sGranB	Nivolumab	[95]
Del Re et al.	2017	KRAS	Erlotinib, Gefitinib	[96]
Dowler Nygaard et al.	2014	cfDNA level, KRAS	Chemotherapy, bevacizumab	[97]
Fiala et al.	2014	NSE, TK	Erlotinib, Gefitinib	[98]
Fiala et al.	2014	CYFRA, CEA	Erlotinib	[99]
Fiala et al.	2015	CRP	Erlotinib	[100]
Haghgoo et al.	2017	TGF-aplha, soluble EGFR, amphiregulin	EGFR TKI	[101]
He et al.	2017	L858R, exon 19 deletion, T790M	Afatinib	[28]
Jiang et al.	2017	MMB	Chemotherapy	[102]
Jiang et al.	2018	CTC count	Afatinib, Erlotinib, Gefitinib, Icotinib	[103]
Juan et al.	2014	CTC count	Chemotherapy	[104]
Karachaliou et al.	2015	Exon 19 deletion, L858R	Erlotinib, chemotherapy	[105]
Lee et al.	2014	BIM	Chemotherapy	[106]
Li et al.	2014	EGFR	EGFR TKI	[37]
Li et al.	2014	RRM1, ERCC1, BRCA1	Chemotherapy	[107]
Ma et al.	2016	EGFR	EGFR TKI	[39]
Mai et al.	2017	CD8/CD4	Montanide ISA 51	[108]
Mok et al.	2015	EGFR	Chemotherapy, Erlotinib	[41]
Muinelo-Romay et al.	2014	CTC count	Chemotherapy	[109]
Nel et al.	2014	CD133/pan-CK, N-cadherin	Chemotherapy	[110]
Nilsson et al.	2016	ALK	Crizotinib	[43]
Nygaard et al.	2014	cfDNA level	Chemotherapy	[111]
Ostheimer et al.	2017	OPN plasma level	Radiotherapy, Chemotherapy	[112]
Oxnard et al.	2016	T790M	Osimertinib	[44]
Paz-Ares et al.	2015	EGFR	Sorafenib	[113]
Qi et al.	2017	CTC count	Chemotherapy	[114]
Que et al.	2016	EGFR	EGFR TKI	[45]
Quoix et al.	2016	TrPAL	TG4010+chemotherapy, chemotherapy	[115]
Shi et al.	2014	survivin mRNA, EGFR	Gefitinib, Erlotinib	[116]
Sun et al.	2017	BIM	Gefitinib, Erlotinib	[117]
Svaton et al.	2014	Natrium level	Erlotinib	[118]
Tissot et al.	2015	cfDNA level	Chemotherapy	[119]
Tu et al.	2017	CD4/CD8, NK expression	EGFR TKI	[120]
Uchibori et al.	2018	Exon 19 deletion, L858R, T790M	Gefitinib+ chemotherapy	[121]
Wang et al.	2017	T790M	EGFR TKI, chemotherapy	[122]
Wang et al.	2018	T790M	EGFR TKI	[123]
Wang et al.	2014	T790M	Gefitinib, erlotinib	[124]
Winther-Larsen et al.	2017	cfDNA mutation	Erlotinib	[125]
Wu et al.	2017	EGFR	Chemotherapy, Afatinib	[126]
Yanagita et al.	2016	CTC count, cfDNA level	Erlotinib	[127]
Yang et al.	2017	CTC count	Gefitinib, Erlotinib	[128]
Yang et al.	2018	CTC count	AZD9291	[129]
Yonesaka et al.	2017	Soluble HRG	Patritumab + Erlotinib	[130]
Zhang et al.	2015	Pokemon	Chemotherapy	[131]
Zhang et al.	2016	sAPE1	Chemotherapy	[132]
Zhou et al.	2017	CTC count	Chemotherapy	[133]
Zhu et al.	2017	EGFR, L858R, exon 19 deletion	EGFR TKI	[134]
Akamatsu et al.	2019	EGFR	Afatinib	[135]
Alama et al.	2019	CTC count, cfDNA level	Nivolumab	[136]
Bordi et al.	2019	Mutation level, EGFR	Osimertinib	[137]
Hojbjerg et al.	2019	miR-30b, miR-30c- miR-211, miR-222	Erlotinib	[138]
Kotsakis et al.	2019	CD4, T-cells, PD-1, PD-L1, B-cells, DC/monocytes	Chemotherapy	[139]
Navarro et al.	2019	Exosome seize	Surgery	[140]
O’Kane et al.	2019	EGFR, T790M	EGFR TKI	[82]
Park et al.	2019	ALK	Crizotinib	[83]
Passiglia et al.	2019	cfDNA level, neutrophil to lymphocyte ratio	Nivolumab	[141]
Tamminga et al.	2019	CTC count	Checkpoint inhibitors	[142]
Wang et al.	2019	TMB	Anti PD-(L)1	[143]
Zhang et al.	2018	Lymphocyte to monocyte ratio, neutrophil to lymphocyte ratio	EGFR TKI	[144]
Yang et al.	2018	MiR-10b	Chemotherapy	[145]
Chea et al.	2019	TMB, MAF	Checkpoint inhibitors	[146]

Cytokeratin 18: CK18, Cytokeratin 19: CK19, Carcinoembryonic antigen: CEA, MET proto-oncogene: c-MET, Granzyme B: sGranB, Kirsten Rat Sarcoma: KRAS, cell-free DNA: cfDNA, Neuron specific enolase: NSE, Thymidine kinase: TK, Cytokeratin-19 fragments: CYFRA, C-reactive protein: CRP, Transforming Growth Factor-alpha: TGF alpha, Epidermal Growth Factor Receptor: EGFR, molecular mutational burden: MMB, circulating tumor cells: CTC, Bcl-2-like protein: BIM, M1 subunit of ribonucleotide reductase: RRM1, excision repair cross-complementation 1 gene: ERCC1, breast cancer susceptibility gene 1: BRCA1, pan-cytokeratin: pan-CK, anaplastic lymphoma kinase: ALK, osteopontin: OPN, triple-positive activated lymphocytes: TrPAL, heregulin: HRG, programmed cell death 1: PD-1, dendritic cells: DC, tumor mutational burden: TMB, mutant allele frequency: MAF.

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
