# Peer review of "The Validity and Predictive Value of Blood-Based Biomarkers in Prediction of Response in the Treatment of Metastatic Non-Small Cell Lung Cancer: A Systematic Review"

_cancers, 2020, doi:10.3390/cancers12051120_

Round 1

Reviewer 1 Report

This review reveals a rage of validity and predict value with liquid biopsy-based biomarkers for patients with NSCLC. Nowadays, this method has spread all over the world. Systemic review has been needed for clinicians.

This systemic review was written well. However, some points need to be reconsidered for sophisticated information.

Major

1) Could you show evidence levels of studies in validity in addition to predictive value?

2) Y-axis is a little bit confusing in Figure 4. CEA? CK19? CK18? If you include these biomarkers such as CEA, CK19 and CK18, only these articles meet criteria for your systemic reviews?

3) Could your show the amount of DNA in NGS or PCR?

Minor

1) “EGFR mutations with exon 19 deletion, L858R, and T790M mutations 98 being were reported in 23, 21, 23, and 10 studies, respectively.” (Line 98) Is this sentence correct?

2) Could the Y-axis in Figure 4 be the same as the Y-axis in Figure 3

3) All references used in this systemic review must be listed.

Reviewer 2 Report

In their manuscript the authors present a systematic review on “The Validity and Predictive Value of Blood Based Biomarkers in Prediction of Response in the Treatment of Metastatic Non-small Cell Lung Cancer”.  This is most certainly a subject that has drawn a lot of attention in the past years. This also becomes clear when seeing the number of hits upon their literature search. Although a very interesting subject for a review, in the end of the day I have to ask myself what I learned from reading it; I failed to see a clear message from this review apart from the conclusion that this is a fast moving field with little concordance between the various studies. It may be inherent to the nature of a systematic review, but I am missing information that I would have expected (hoped, maybe) to be included.

The large amount of variation in the methodologies (also mentioned in l171-174) in this field actually appears to hamper the impact of a systematic review as presented here.

Major comments:

Overall there is little attention for the specific methodology that is used in the various studies. This is especially problematic as it is almost impossible to trace back the specific papers on which the presented data are based. Even within the group of studies that have used PCR, there may be variation in the methodology. The same will be true for the NGS group.  Different methodologies may result in different sensitivities.  A related issue: it has not been stated if - in the papers used for this review – the methodology was the same for the LB and TB samples. Nor is it discussed if they even should be the same. Maybe a different source of material warrants a different type of analysis for optimal sensitivity. All these points appear to limit the informativity of Fig 2.

This review would gain a lot in informativity if the authors would have given more information on the methodology, especially for those studies that show the highest sensitivity/specificity/concordance.

The references should be included in Fig 2 and Fig 4. For Fig  2 this appears to be difficult, but could maybe be accomplished using a supplementary Fig. I now cannot match the sensitivity data with the specificity data and/or the concordance data. Also, it is almost impossible to trace back the specific publication that is underlying any of the spots in Fig 2 or any of the studies presented in Fig 4.

Fig 2 raises a lot of questions. The listed mutations are not well explained. For instance: what does EGFR-PCR mean? For the sensitivity track: The blocks are explained as the range, but for several tests I see blue spots outside the range. It is not clear how the range is calculated, and thus what this range represents. Do the authors mean the 25-75% range? As an example, for EGFR_PCR I see several large spots outside the block that indicates the range. For L858R_cSMART the blue spots appear to be missing, or are at least not visible. In several cases throughout Fig 2 the data appear to be extracted from only one study. In these cases it is often impossible to see how large this study was. The red crosses give the mean sensitivity for each group of assays. But I am not sure if I this is a meaningful value given the large variation in the methodology. I would be much more interested in the study with the highest sensitivity, and the methodology applied. As one out of several examples, when looking at the sensitivity for the L858R assays, NGS and PCR perform equal when looking at the mean. But there are PCR studies that outperform almost all NGS studies.

Fig 4: I am not familiar with the evidence framework as proposed by Rao. I assume that the top block (1D) represents the most trustworthy studies. However, I would have expected to also see something like a p-value, or a sensitivity value that would indicate the strength of the correlation of the observed variant with the outcome of the therapy.

Minor comments:

L90 says that studies had to include al least 10 patients. But in Fig 1 the publications were already filtered on having at least 20 patients. Please explain

Fig 2, and l106 and l109: It is unclear what is meant by the indication EGFR (as in EGFR_PCR). 

Fig 2: I wonder how the authors have decided on the specific layout, e.g. the ranking of the various tests. Why not group all the EGFR assays, and/or group all non-DNA based assays. The ranking now seems random.

L97/98 this sentence appears to be on the wrong position in the text.

L100: This is an unexpected observation. However, there are many ways to do PCR. All with different types of sensitivity. The same holds true for NGS. So this conclusion is not very informative without further detailed information about the methodology.

L101-104: this information should be included in the legend of Fig.2

L116-123: in the discussion on the concordance I am missing a discussion on the possible explanation of these findings. Especially since one would assume a (dd)PCR-based assay in general to be more sensitive as NGS. Did any of the studies report on the concordance between PCR and NGS?

Fig 4: the colors are not always easy to interpret. For instance the color for miR-221. Does it refer to gefitinib or erlotinib?

L180-l181: The authors argue that – based on the observed sensitivity - LBs are not ready to replace tissue biopsies in practice. However, one could argue that the study showing the best sensitivity and/or specificity should in fact be leading – and not the overall spread in the data.

L182-184. Should we take into account that also LB may be biased as it may predominantly be derived from apoptotic cells, and thus not every subpopulation of tumor cells is equally represented.

L184. IASLC should be written in full.

Supplementary data: I do not see the added value of having three independent reference lists. I would prefer to have one, with references also added to Fig 2 and Fig 4.

Reviewer 3 Report

The role of liquid biopsy to assess potential biomarkers for NSCLC represents a very broad and growing research field. The number of reviews in the scientific literature has exponentially increased in the last years. In order to be of use to the readers, a review should be critical and should contribute to this crowded area.

I appreciated the review methods but I would prefer them before the results and discussion sections.

The discussions should better elucidate the meaning and clinical impact of this work. Unfortunately, this review is a little contribution to the research field.

As the authors admitted, most studies focused on detection EGFR mutations. In particular, liquid biopsy was investigated for the detection of the T790 mutation as a mechanism of resistance to first and second generation TKI when these were the standard of treatment for EGFR mutated NSCLC. The T790M mutation is predictive biomarker of response to Osimertinib, a third generation TKI. This represents the main application of the liquid biopsy approved by the FDA in 2016.

The authors should be more critical considering two very important aspects:

  1. Lung tumors with EGFR mutation represent a minority of cases
  2. Osimertinib has become the new standard of care as frontline treatment for EGFR-mutant NSCLC. This has reduced the use of liquid biopsy in clinical practice.

The authors should declare these “limitations” and emphasize the potential of the liquid biopsy to identify other biomarkers, certainly less studied than EGFR but with growing interest and with greater promising applicability in future clinical practice. 

Round 2

Reviewer 1 Report

This paper is well-revised.

Reviewer 2 Report

The authors have more than adequately responded to the many remarks that I made reviewing their paper. 

They have now included all the references in the manuscript and added two tables to make it easier to trace back the papers that ended up in Fig 2 and Fig 4. For Fig 4, this works perfectly. For Fig 2 it is acceptable, although still not perfect.

There is still one issue on which we now “agree to disagree”.  One of my remarks concerned that there is a lot of variation in PCR and NGS protocols and this makes it difficult to make generalized comments about the performance of PCR in comparison to NGS. However, the authors have addressed this issue in the discussion, where they explain their choice on this matter. Although I accept the choice the authors made, I think it does limit the impact of this paper. 

Having read the new version of the manuscript, I have no further comments. 

Reviewer 3 Report

I believe the manuscript has been significantly improved

In my opinion, no other changes are required 

Thank you